# The Effectiveness of a Recreational Behavioural Programme in Reducing Anger among Children with Intellectual Disabilities at the Primary Stage

Ahmed R. Elsayed [1,*] and Ahmed K. Hassan [2,3]

1    Department of Special Education, College of Education, King Faisal University, Al-Ahsa 31982, Saudi Arabia
2    Department of Physical Education, College of Education, King Faisal University, Al-Ahsa 31982, Saudi Arabia;
     amohhamed@kfu.edu.sa
3    Department of Team Sports and Racket Games, Faculty of Physical Education, Minia University,
     Minya 61519, Egypt
*    Correspondence: arelsayed@kfu.edu.sa

**Abstract:** This study aimed to investigate the impact of a recreational behavioural programme on reducing the degree of anger among children with intellectual disabilities (ID) in the primary stage. The study was implemented with 24 children who were randomly divided into two groups: an experimental group (n = 12, age = 10.80 ± 1.03 years, IQ = 63.10 ± 4.43 scores, ASW = 55.50 ± 1.51 scores) and a control group (n = 12, age = 10.80 ± 0.92 years, IQ = 63.00 ± 4.16 scores, ASW = 56.00 ± 1.15 scores). We used the PROMIS anger scale with a modification that was used to measure the degree of anger, and the recreational behavioural programme was implemented three times per week for six weeks. The results of the research showed that the improvement percentages for Anger Triggers (AT), Inner Anger (IA), and External Anger (EA) were 9.73%, 9.04%, and 9.60%, respectively, and the Anger scale as a whole (ASW) rate was 9.46%. r = (0.89–0.91). The experimental group using the recreational behavioural programme also outperformed the control group, as the results indicated a decrease in the intensity of anger in the direction of the experimental group. The differences in the improvement percentages for Anger Triggers (AT) Inner Anger (IA), and External Anger (EA) were 32.97%, 31.03%, and 26.63%, respectively, and the Anger scale as a whole (ASW) rate was 30.09%, r = (0.82–0.86). The results of the study confirmed the effectiveness of the recreational activity programme in developing social interaction among children with intellectual disabilities, which indicates the success of the recreational behavioural programme in reducing the degree of anger among children with IDs. Therefore, the recreational behavioural programme had a positive effect in terms of reducing the degree of anger among children with IDs in the primary stage.

**Keywords:** PROMIS anger scale; anger stimuli; internal anger; external anger; intellectual disabilities (IDs); intelligence (IQ)

## 1. Introduction

Anger is a negative affective state that may include increased physiological arousal, thoughts of blame, and an increased predisposition towards aggressive behaviour [1]. Intense and out-of-control outbursts of anger may be of clinical concern in young children [2,3]. Intense outbursts of anger in response to trivial provocations may also persist across developmental stages and manifest in various psychiatric disorders. Because of an apparent lack of control, these behaviours have been referred to as "rage attacks" in severe mood dysregulation [4]. Irritability is described as having a low threshold for experiencing symptoms of negative affectivity, such as anger, in response to frustration. Thus, anger, frustration, aggression, and irritability are interrelated psychological constructs [5].

Children with IDs show elevated rates of both internalising and externalising problem behaviours compared to typically developing children [6,7]. Anger regulates personal and

social behaviours, and the inappropriate expression and regulation of anger have social and clinical implications [8,9]. In this regard, Deno et al. [10] indicated that the expression of anger was related to children's psychological and social adjustment as well as their mental health. Kaman et al. [11] stated that anger was a common symptom in children and caused numerous problems for them; thus, it is necessary to provide therapeutic interventions to reduce these children's levels of anger. In addition, this was confirmed by Nicoll and Beail [12] in their study of the presence of high levels of anger in children with IDs. Children with mental disabilities have low levels of compatibility between the content of their knowledge of normal relationships and the dispensed procedures. Thus, these children have an inaccurate and inappropriate level of cognitive performance from a constructive perspective [13]. with low self-esteem and increased emotional sensitivity, which leads to their inability to exercise psychological control as well as self-regulation [14]. In addition, Hamelin et al.'s [15] study indicated the possibility of treating anger in children with IDs. In this regard, Travis and Sturmey [16] studied the treatment of anger in people with IDs via training in behavioural skills, which proved to be successful in treating anger in children with IDs, as well as increasing their social adaptation and removing some of their negative behaviours. The study by Borji et al. [17] concerned the need for the motor rehabilitation of children with IDs due to its important effects on increasing their strength as well as improving their motor and physical abilities.

Roberts et al. [18] confirmed the effectiveness of behavioural interventions in treating anger in children and stated that behavioural interventions also played a major role in treating anger in children with IDs. McGuire and McDonnell [19] found that recreation had positive results for individuals with IDs and that giving these individuals opportunities to engage in recreational activities had a positive effect on them in terms of social and emotional aspects. Recreation is defined as a term that includes activities that contain physical activity as well as other activities that are practised independently in open spaces [20]. With regard to anger management, CBT in the general literature has a growing evidence base that has been reported in several meta-analyses [21,22]. "It has been noted that treatment for people with intellectual disabilities has been absent from "what works for whom"" [23]. The literature reviews indicate a lack of research on the effectiveness of psychotherapy applied to this population [24]. Whitaker [25] concluded that the evidence for CBT as an effective treatment for anger in people with intellectual disabilities was weak, and more recent narrative reviews suggest an emerging evidence base for the effectiveness of anger CBT interventions in those with intellectual disabilities [26,27]. The treatment of anger issues has become one of the most widely researched issues in the field of intellectual disabilities [26–28]. Recreational activities are important for individuals who suffer from psychological and behavioural problems, because recreation has the tremendous benefit of allowing individuals to vent their emotions in positive ways by engaging them in activities that they love and creating an atmosphere of joy [29]. As Hian et al. [30] pointed out, it is important for individuals to engage in recreational activities in the open air due to their benefits in terms of improving their psychological and physical health. In turn, this indicates the importance of recreational activities for children as well as the importance of providing various forms of recreation for them, ranging from parks to a variety of activities [20]. In this regard, Curtin et al. [31] emphasised the need to provide physical and recreational activities for people with IDs, as with other individuals, and to give them the opportunity to practise physical, motor, and recreational activities because of their benefits in various psychological, social, and physical fields. Participation in leisure activities is a prerequisite for human development and has the potential to satisfy the basic social and psychological needs of people with disabilities [32].

It is clear from the foregoing that the previous treatments may not be completely effective in treating or reducing anger in children with intellectual disabilities and that no research has analysed their effects on anger; this is what prompted us to conduct the study. The study is also important for those working in the field of special education, as well as those concerned with children with a personal identity and their families. Accordingly,

this study aimed to reveal the most important aspects of anger in children with IDs that need to be addressed. In order to confirm the effectiveness of the recreational behavioural programme in reducing anger in children with IDs, the present study attempted to determine the extent to which the proposed recreational behavioural programme contributed to decreasing anger among children with IDs. We assumed that there would be statistically significant differences between the mean scores of the experimental and control groups for children with personal identity on the scale of anger in the scores of the premeasurement, followup measurement, and post-measurement in favour of the post-measurement scores of the experimental group.

## 2. Materials and Methods

### 2.1. Participants and Study Design

The study programme was implemented over a period of six weeks, with three sessions each week, and this is shown in Table 1. it included children with identity cards (ID) with intellectual disabilities, who were able to learn, including children with Down syndrome. We became acquainted with their social conditions and their detailed data through the children's records who were capable of learning and who were enrolled in intellectual education classes at the primary stage in Al-Ahsa Governorate, Saudi Arabia. The children of the study sample belonged to one social and economic level, which was the above-average social level, and this is shown in Table 2. The programme was implemented for a sample of 24 children who were randomly divided into two groups: an experimental group (n = 12, age = $10.80 \pm 1.03$ years, intelligence = $63.10 \pm 4.43$ scores, ASW = $55.50 \pm 1.51$ scores) and a control group (n = 12, age = $10.80 \pm 0.92$ years, intelligence = $63.00 \pm 4.16$ scores, ASW = $56.00 \pm 1.15$ scores). The results revealed that there were no differences between the two samples, which means that the two groups were equal. With regard to the research variables, the Stanford–Binet Intelligence Scale was used. The researchers supervised the recreational behavioural programme, and the results of the measurements were compared before, during, and after the application. We used the primary data form for children with ID to obtain data related to children with ID in an overlapping and accurate manner in order to choose a similar sample of children (see Appendix A).

**Table 1.** Recreational behavioural programme sessions.

| Session Number | Session Topic | Session Duration | Techniques and Activities Used | Number of Sessions |
|---|---|---|---|---|
| First | Acquaintance and introduction to the program. | 30 min | Strengthening | 1 |
| Second | Definition of anger and why I get angry (causes of anger). | 30 min | Reinforcement and modelling | 1 |
| Third and fourth | The consequences of anger and its impact on others' acceptance of us. | 30 min | Augmentation, modelling, and role playing | 2 |
| Fifth and sixth | How do I control the emotion of anger? | 30 min | Augmentation, modelling, and role playing | 2 |
| Seventh–Eleventh | Apply the response cost action. | 30 min | Boosting, modelling, role playing, and response cost | 5 |
| Twelfth | Recreational sports activity (football). | 30 min | Reinforcement, football | 1 |
| Thirteenth | Recreational sports activity (basketball). | 30 min | Reinforcement, basketball | 1 |
| Fourteenth | Recreational sports activity (tug of war). | 30 min | Reinforcement, tug of war | 1 |
| Fifteenth | Recreational sports charts (jumping inside hoops). | 30 min | Boosting, jumping inside hoops | 1 |
| Sixteenth | Closing session. | 30 min | Strengthening | 1 |

**Table 2.** The homogeneity of the study sample in terms of the social and economic level.

| Group | N | Mean Rank | Sum of Ranks | Z |
|---|---|---|---|---|
| experimental | 12 | 10.65 | 106.50 | 0.92 |
| control | 12 | 10.35 | 103.00 | |

*2.2. Ethical Considerations*

The parents were fully informed about the risks and benefits of the study prior to the trial, and signed parental consent for children with IDs was obtained. The protocol was approved by the Research Ethics Committee of King Faisal University, Saudi Arabia (KFU-REC-2022-DEC-ETHICS370).

*2.3. Assessments*

Assessments were conducted at baseline and before, during, and after the intervention, respectively, in order to investigate the effectiveness of a recreational behavioural programme in reducing anger in children with intellectual disabilities at the primary level. We used tools with training in their use, which were presented to people with experience, and there were levels of agreement among the raters [statistic $\geq 0.96$]. The following tools were selected as a raw data model for children with IDs: the Anger Scale (see Appendices A–C for details). We used the PROMIS anger scale [11] with a modification that was used to measure the degree of anger in a sample of children with IDs. In its initial form, the scale contained 22 items that were distributed across the three dimensions of the scale: anger triggers (AT), inner anger (IA), and external anger (EA). The scale items were distributed with seven items for each of the two dimensions, AT and LT, and eight items for the EA dimension. The validity of the scale was verified by presenting it to a group of teachers of special education and mental health, and their comments on the scale were taken into consideration. The correlation coefficients were calculated between the degrees of the aspects of each dimension and the degree of the dimension to which they belonged, and the correlation coefficients ranged between (0.654 and 0.925) for all three dimensions of the scale. Calculating the correlation coefficients between the degree of each dimension of the scale and the total score of the scale ranged between (0.782 and 0.920). The reliability coefficient of the scale was calculated by two half-partition methods, the Spearman–Brown equation (AT) (0.955), (IA) (0.889), (EA) (0.873), and (ASW) (0.915) and Cronbach's alpha (AT) ((0.906), (IA) (0.908), (EA) (0.886), and (ASW) (0.926). The researchers measured the scale with children with intellectual disabilities in Al-Ahsa, Saudi Arabia. After the scale rating process, the final form of the scale included 21 vocabulary items, which were distributed across the three main dimensions of anger, anger triggers (AT), inner anger (IA), and extrinsic anger (EA), with seven items for each dimension. The semi-experimental approach was used, and the experimental design with two experimental and control groups was chosen.

Testing Procedures

Before starting the programme, the two research groups were pretested on 13 and 14 September 2022. The researchers aimed to apply the tests to all individuals in a standardised manner. The Recreational Behavioural Programme was implemented for a period of six weeks, starting on 18 September 2022 and ending on 27 October 2022, with three sessions per week on Sundays, Tuesdays, and Thursdays. After completing the implementation of the programme, the telemetry was conducted from 30 to 31 October 2022, using the same method that was followed in the premeasurement stage and under the same conditions.

*2.4. Recreational Behavioural Program*

This programme, which was one of the fundamental tools created to accomplish the goals of the current study, made use of a variety of behavioural techniques, including reinforcement, modelling, role playing, and response cost, as well as leisure pursuits

such as sports, artistic endeavours, and social interactions, with the goal of determining their contribution to lowering the level of anger among children with IDs. Through the recreational behavioural plan, the programme aimed to lessen the anger of children with IDs. The six-week recreational behavioural programme, which had 18 sessions (three per week), was put into practise in Al-Ahsa Governorate. Each session lasted 30 min. Depending on each person's needs and the focus and character of each session, we used both group and individual counselling. The programming and the session's material were explained to a few teachers who had the training to handle these situations. Beginning with the second semester of the academic year 2022–2023, the programme was put into effect. Three levels of assessment were used to assess the programme: the interim evaluation at the end of each session of the programme, the post-evaluation, and the followup evaluation at the conclusion of each session. Three weeks after the programme's implementation, the experimental group's participants had a tracking measurement of the anger scale. Table 1 displays the breakdown of the six-week recreational behavioural programme sessions by the approaches and activities applied to the experimental group, as well as the session duration.

*2.5. Statistical Analysis*

Since the category of children with intellectual disabilities (children with Down syndrome), to which the research was applied, is a small population that accounts for only approximately 10% of all children with intellectual disabilities, a small sample of children was used.

For the statistical package, the social sciences statistical programme (SPSS; IBM SPSS Statistics 26.lnk, Chicago, IL, USA) was utilised. The Spearman–Brown equation, along with the mean, standard deviation, Z value, effect size (r), and percentage, were used. With a *p* value of 0.05, the findings were deemed statistically significant. In order to determine the reliability and validity of the anger scale, Cronbach's alpha was used. The significance of the change in the rankings of the members' scores was assessed using the nonparametric Wilcoxon test before and after, post-test and followup, in the related (experimental) groups, and the nonparametric Mann–Whitney U test was used to determine the significance of the variations in the mean scores of the participants in the independent groups (experimental and control) before, during, and after assessment.

**3. Results**

For the experimental and control groups, the data from the pre–post and tracking measurements, as well as the statistics for the children's scores on the anger scale, are given. Table 2 demonstrates the social and economic level similarities of the study population. The scores of the experimental group's pre- and post-measurements on the anger scale changed in the direction of the post-measurement, as shown in Table 3. $p = 0.005$, r = (0.89–0.81), respectively. In Table 4, it is shown that the scores of the control group's pre- and post-measurements on the anger scale differed in the direction of the post-measurement ($p = 0.005$, r (0.82–0.86)). Additionally, Table 5 revealed that there were disparities between the experimental and control groups' post-measurements on the anger scale, with the experimental group showing the most variances (r = (0.96–1.24). According to Table 6, there were no changes between post and tracking, with r = (0.55–1.73).

Table 2 clearly shows that the value of (Z) for the two research groups was not statistically significant, demonstrating that the study sample was homogeneous in terms of the social and economic status.

Figures 1–3 (the anger scale for children with IDs) display the rankings for the two groups as well as the Z values for the prior and following measurements of the experimental and control groups, while Figure 4 displays the rankings for the experimental group for the inter-measurement and post-measurement scores on the dimensions of the anger scale.

**Table 3.** The importance of the variations in the experimental group's mean scores between the pre- and post-tests on the rage scale.

| Variables | | M | SD | Z | r | Percentage % | Sig |
|---|---|---|---|---|---|---|---|
| AT | Pre | 18.20 | 1.03 | 3.111 | 0.90 | 32.97% | 0.002 |
| | Post | 12.20 | 0.63 | | | | |
| IA | Pre | 17.40 | 0.84 | 3.169 | 0.91 | 31.03% | 0.002 |
| | Post | 12.00 | 0.67 | | | | |
| EA | Pre | 19.90 | 0.57 | 3.130 | 0.90 | 26.63% | 0.002 |
| | Post | 14.60 | 0.70 | | | | |
| ASW | Pre | 55.50 | 1.51 | 3.076 | 0.89 | 30.09% | 0.002 |
| | Post | 38.80 | 1.55 | | | | |

AT = anger triggers; IA = inner anger; EA = external anger; ASW = anger scale as a whole; Sig = statistical significance; Z value at (0.05) = 1.96, at (0.01) = 2.58.

**Table 4.** The importance of the variances in the control group's mean scores between the pre- and post-tests of the anger scale.

| Variables | | M | SD | Z | r | Percentage % | Sig |
|---|---|---|---|---|---|---|---|
| AT | Pre | 18.50 | 0.97 | 2.972 | 0.86 | 9.73% | 0.003 |
| | Post | 16.70 | 0.82 | | | | |
| IA | Pre | 17.70 | 0.67 | 2.889 | 0.83 | 9.04% | 0.004 |
| | Post | 16.10 | 0.57 | | | | |
| EA | Pre | 19.80 | 0.63 | 2.850 | 0.82 | 9.60% | 0.004 |
| | Post | 17.90 | 0.57 | | | | |
| ASW | Pre | 56.00 | 1.15 | 2.831 | 0.82 | 9.46% | 0.005 |
| | Post | 50.70 | 1.06 | | | | |

AT = anger triggers; IA = inner anger; EA = external anger; ASW = anger scale as a whole; Sig = statistical significance; Z value at (0.005) = 1.96, at (0.01) = 2.58.

**Table 5.** The importance of the variances in the mean scores of the two post-tests on the anger scale for the children in the experimental and control groups.

| Variables | Group | N | M | SD | Z | r | DIR % | Sig |
|---|---|---|---|---|---|---|---|---|
| AT | Experimental | 12 | 12.20 | 0.63 | 4.271 | 1.23 | 23.24% | 0.000 |
| | control | 12 | 16.70 | 0.82 | | | | |
| IA | Experimental | 12 | 12.00 | 0.67 | 4.281 | 1.24 | 21.99% | 0.000 |
| | control | 12 | 16.10 | 0.57 | | | | |
| EA | Experimental | 12 | 14.60 | 0.70 | 3.322 | 0.96 | 17.03% | 0.000 |
| | control | 12 | 17.90 | 0.57 | | | | |
| ASW | Experimental | 12 | 38.80 | 1.55 | 4.203 | 1.21 | 20.63% | 0.000 |
| | control | 12 | 50.70 | 1.06 | | | | |

AT = anger triggers; IA = inner anger; EA = external anger; ASW = anger scale as a whole; DIR = differences in improvement rates; Sig = statistical significance; Z value at (0.05) = 1.96, at (0.01) = 2.58.

**Table 6.** The importance of the variations in the experimental group's mean scores between the tracking and post-measurements of their anger.

| Variables | | M | SD | Z | r | Percentage % | Sig |
|---|---|---|---|---|---|---|---|
| AT | Tracking | 12.50 | 0.85 | 1.732 | 0.55 | 2.40 | 0.083 |
| | Post | 12.20 | 0.63 | | | | |
| IA | Tracking | 12.10 | 0.74 | 1.000 | 0.58 | 0.83 | 0.317 |
| | Post | 12.00 | 0.67 | | | | |
| EA | Tracking | 14.80 | 0.92 | 1.414 | 1.41 | 1.35 | 0.157 |
| | Post | 14.60 | 0.70 | | | | |
| ASW | Tracking | 39.40 | 1.84 | 2.449 | 1.73 | 1.52 | 0.014 |
| | Post | 38.80 | 1.55 | | | | |

AT = anger triggers; IA = inner anger; EA = external anger; ASW = anger scale as a whole; Sig = statistical significance; Z value at (0.005) = 1.96, at (0.01) = 2.58.

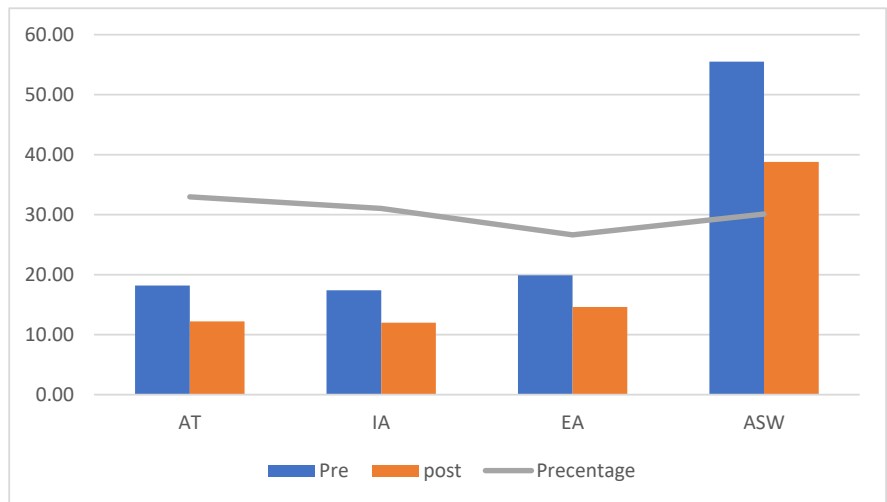

**Figure 1.** Differences between the mean scores of the pre- and post-measurement scores in the anger scale for the children in the experimental group.

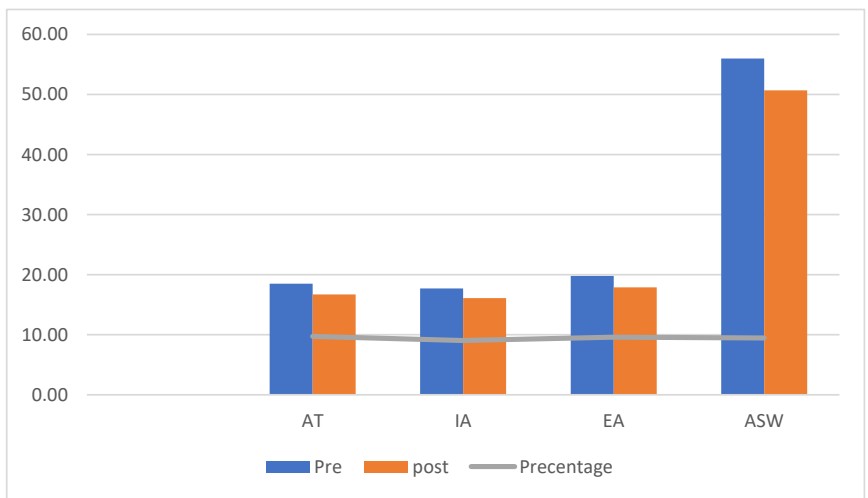

**Figure 2.** Differences between the mean scores of the pre- and post-measurement scores in the anger scale for the children in the control group.

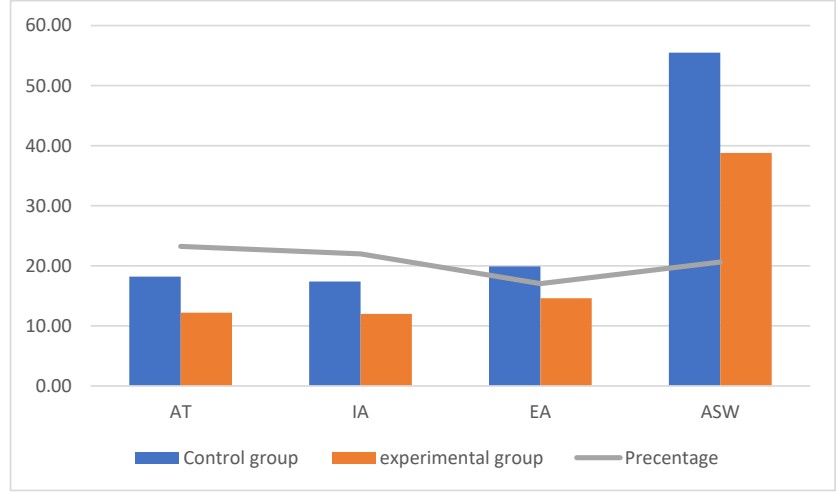

**Figure 3.** Differences between the mean scores of the post-measurement scores on the anger scale for the children in the experimental and control groups.

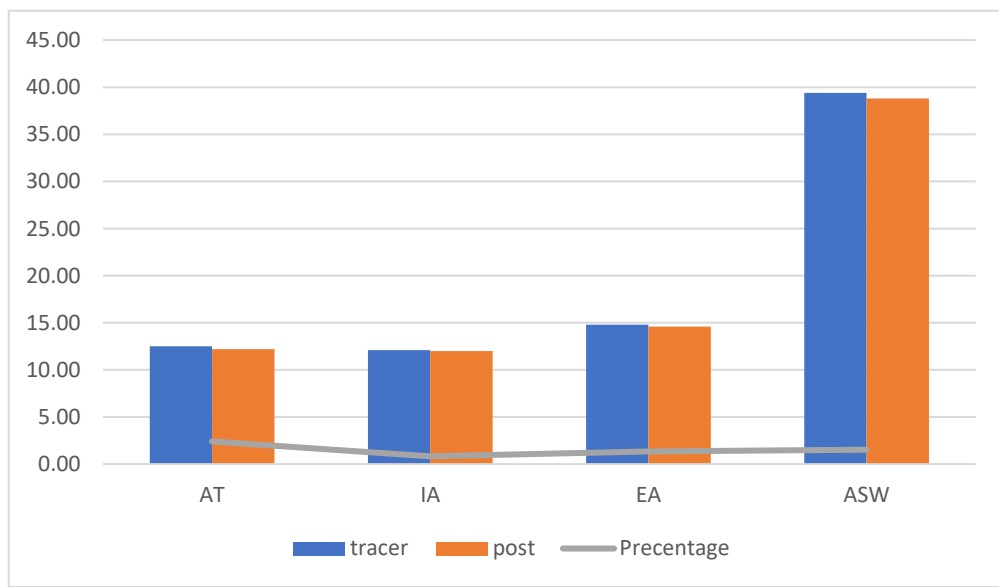

**Figure 4.** Differences between the mean scores of the tracking and post-measurement scores on the anger scale for the children in the experimental group.

We found a decrease in the intensity of anger in the post-measurement in the dimensions of the scale for the AT, IA, EA, and ASW [*p* = 0.05], as shown in Figure 1, which illustrates the significance of the differences between the mean scores of the pre- and post-measurement scores on the anger scale for the children in the experimental group. Table 3 shows the differences between the mean scores of the pre- and post-measurement scores in the anger scale for the children in the experimental group.

Table 3 displays statistically significant differences between the pre- and post-measurement anger scale ratings for the experimental group, since the calculated Z value was higher than the tabular Z value. *p* = 0.05. This was due to the anger scale's significant effect size (r).

The control group's mean scores on the anger scale varied significantly between pre- and post-measurement, as shown in Figure 2. This finding suggests that the post-measurement anger intensity was lower in the AT, IA, EA, and ASW. The significance of the variations in the mean scores of children in the control group and the Z value for the dimensions on the anger scale for children with IDs are shown in Table 4.

The estimated Z value was higher than the Z value of the tabular *p* = 0.005; the effect size (r) on the anger scale had a substantial effect size, and Table 4 displays the statistically significant differences between the pre- and post-measurement scores on the anger scale for the control group.

The average post-measurement scores on the anger scale for the children in the experimental and control groups were significantly different, as shown in Figure 3. This finding suggests that the experimental group saw a decrease in the intensity of their anger. Additionally, we discovered that the rates of improvement in the AT, IA, EA, and ASW anger scale dimensions varied. The Z value in the dimensions on the anger scale for children with IDs is shown in Table 5, together with the significance of the differences between the mean scores of the children in the experimental and control groups.

Because the computed Z value was higher than the Z value in the table and because the effect size (r) on the anger scale had a large effect size, Table 5 displays the statistically significant differences between the experimental and control groups in the dimensional measurements of the anger scale.

The experimental group's mean tracking and post-measurement scores on the anger scale did not differ significantly, as shown in Figure 4. This suggests that we found relatively low scores for the severity of anger in the direction of the post-measurement scores in the dimensions of AT, IA, EA, and ASW of the anger scale. The Z value of the dimensions on the anger scale for children with IDs is presented in Table 6, along with the

significance of variations between the tracking and post-measurement scores for children in the experimental group.

Table 6 shows that there were no statistically significant variations between the tracking and post-measurements for the experimental group's anger scale, as the computed Z value was less than the Z value and $p > 0.05$, except for the (ASW) $p = 0.014$, due to the anger scale's effect size (r) having a significant effect size for the post-measurement.

## 4. Discussion

The purpose of the study was to determine whether a recreational behavioural programme affected how much rage primary-stage ID children displayed. The significance of the differences between the average ranks of the degrees of measurements for the experimental group and the control group before and after the programme's implementation in the dimensions on the anger scale, as well as the total degree, were examined by the researchers using the ranks of the signals of the two groups. The results showed a drop in the children's levels of anger in the experimental group. We discovered that the experimental research group's levels of anger were lower in the post-measurement than in the pre-measurement. The results showed that the children in the experimental group had less intense anger. In the experimental research group, where the improvement percentages for AT, IA, and EA were 9.73%, 9.04%, and 9.60%, respectively, and the ASW rate was 9.46%, we discovered a decrease in the intensity of anger between the pre- and post-measurements. Because the results showed that the experimental group's anger was less intense towards the experimental group, they also outperformed the control group when utilising the recreational behavioural programme. The ASW rate was 30.09%, while the variances in the improvement percentages for the AT, IA, and EA were 32.97%, 31.03%, and 26.63%, respectively. We discovered that the value of (r) varied between 0.96 and 1.24 in this case. It had a significant impact because the behavioural strategies and leisure activities used with the experimental group of children with IDs were successful in reducing the level of anger as measured by the dimensional scales. The members of the experimental group regularly attended the sessions and activities in the programme used in the study. The behavioural techniques and recreational activities used were significant in the members' lives, giving them the chance to gain from the programme's techniques and activities within the context of enjoyable recreational activities.

This helped them feel less angry overall. Additionally, the effectiveness of the programme was determined by tracking the children's behaviour using tracking measurements as well as feedback from parents and instructors. As a result, we concluded that the children's leisure activities had a good effect. According to Travis and Sturmey's [16] research, training in behavioural skills can effectively manage anger in people with intellectual disabilities, as well as improve their social adaptability and reduce some of their negative behaviours. The study by Borji et al. [17] showed a significant impact on boosting their strength and enhancing their physical and motor skills. In addition to stating that behavioural therapies were extremely important in treating anger in children with IDs, Roberts et al. [18] also confirmed the efficacy of behavioural interventions in treating anger in children. According to McGuire and McDonnell [19], giving people with IDs opportunities to engage in recreational activities has a good impact on them in terms of social and emotional components. Recreation has favourable results for people with IDs. The fact that the children's emotions may be conveyed in appropriate ways highlights the value of and the need for promoting physical activity and leisure pursuits for those with IDs [33–35]. The reduced levels of anger among the children with IDs in the experimental group demonstrated the success and viability of executing the leisure activities included in the programme. Through their involvement in tug-of-war games and running races, in particular, sporting activities helped these children's levels of anger decrease.

These activities allowed them to express their rage by engaging in the program's motor activities, which helped them release their wrath. Additionally, the children's participation in team activities on a daily basis [36] allowed them the chance to use their energy and express their emotions in constructive ways. By fostering an atmosphere of joy and pleasure and allowing the children to express their angry emotions through the artistic recreational activities they engaged in during the programme sessions, the artistic activities also played a crucial role in lowering the level of anger among the children with IDs in the experimental group. Children's fitness and mental health are greatly influenced by physical activity. Additionally, children who are in good mental health are more likely to participate actively in society [37,38]. The prosocial behaviours of children with special needs can be enhanced by encouraging them to engage with a variety of people and actively explore different life skills, such as teamwork [39], problem solving [40], and goal setting [41]. Social activities also played a clear role in decreasing the levels of anger of the children in the experimental group; some of the social activities included in the programme were competitions and recreational games, which gave the children the opportunity to abandon their emotions of anger and to replace them with social behaviours, to make new friends with children through working groups formed by children, and to develop the ability to understand others; these results are consistent with the results of other research and studies that used recreational activities to reduce the level of anger in children with IDs [17,31], as these studies proved the effectiveness of recreational activities in reducing the degree of anger in children in general and among those with IDs in particular. This was in addition to the role played by the techniques in reducing the degree of anger in the members of the experimental group.

Furthermore, the results indicated the continued effectiveness of the recreational behavioural programme after a month of implementation with the children in the experimental group in reducing the level of anger they had, which the researchers believe was the result of the exposure to natural light in the recreational activities included in the programme as well as the interaction, cooperation, and commitment shown by the children during the programme sessions. This result was expected due to the results of previous studies that proved the effectiveness of behavioural techniques and recreational activities in the ongoing reduction in the degree of anger and some unwanted behaviours in children with IDs [6,16,17,22,31–33,36]. In conclusion, it was discovered that the programme, which included a variety of enjoyable activities, helped children with IDs control their anger. The effectiveness and viability of the behavioural techniques and recreational activities demonstrated their value in providing children with IDs the chance to modify their angry behaviour and to discharge their angry emotions in acceptable ways through participation in activities they love, such as the sports and the artistic and social recreational activities included in the programme, under the direction of researchers.

*Study Limitations and Strengths*

The study studied children with Down syndrome, where the scale was applied to them, and the first limitation was that the study did not address other disabilities, for example, cerebral palsy. The second limitation was related to the relatively small sample size; therefore, the results are not generalizable to other regions and institutions. The third limitation was not examining the effect of the behavioural entertainment programme on reducing the anxiety and stress in children with IDs. The participants were recruited via convenient sampling rather than random sampling. The strengths of the study were represented by the contribution provided by the programme in reducing anger in the children with IDs, which contributed to reducing the burden on the parents of these children. Therefore, because of the above, intellectual education classes must provide sports and recreational behavioural programmes to reduce the phenomenon of anger in these children. Applying the behavioural entertainment programme to other samples, other disabilities, and different age stages is needed.

## 5. Conclusions

In conclusion, it was discovered that the programme, which included a variety of enjoyable activities, helped children with IDs control their anger. The effectiveness and viability of the behavioural techniques and recreational activities demonstrated their value in providing children with IDs the chance to modify their angry behaviour and to discharge their angry emotions in acceptable ways through participation in activities they love, such as the sports and the artistic and social recreational activities included in the programme, under the direction of the researchers. Children who become angry may have difficulty adjusting to their social surroundings. This is in line with several other research and study findings [9,11,12,42–44]. According to these results, children with IDs exhibited anger-related feelings and needed interventions to lessen their severity. Aggression and intense, frequent, and uncontrollable anger have a negative impact on both the individual and the society in which they live because uncontrolled anger encourages aggression and violent behaviour [22,24,27,45]. Aggression and uncontrollable anger should be reduced in order to help people change their actions and behaviours towards others and improve their social compatibility. In order to improve generalizability, future studies should strive to include more individuals with intellectual disabilities. To lessen children's aggression, we advise certifying counsellors and giving them training in behavioural recreational programmes. It is in the best interests of specialised institutions to encourage the implementation of these programmes in order to lessen aggressive behaviour and rage, as well as foster a sense of community and collaboration among these children.

**Author Contributions:** Conceptualization, A.R.E. and A.K.H.; methodology, A.R.E.; formal analysis, A.K.H.; investigation, A.K.H. and A.R.E.; resources, A.R.E. data curation, A.K.H.; writing—original draft preparation, A.R.E. writing—review and editing, A.R.E. and A.K.H.; visualization, A.R.E.; supervision, A.K.H. project administration, A.R.E. and A.K.H.; funding acquisition, A.R.E., All authors have read and agreed to the published version of the manuscript.

**Funding:** This study was funded by the Deanship of Scientific Research at King Faisal University, Saudi Arabia, grant number (GRANT1,807).

**Institutional Review Board Statement:** This study was conducted according to the guidelines of the Declaration of Helsinki and approved by the Research Ethics Committee at King Faisal University (protocol code KFU-REC-2022-DEC-ETHICS370, approved on 6 December 2022).

**Informed Consent Statement:** Informed consent was obtained from all subjects involved in the study.

**Data Availability Statement:** Data are contained within the article.

**Acknowledgments:** The authors wish to thank all the subjects who participated in this study. Informed consent was obtained from all study participants, including their parents.

**Conflicts of Interest:** The authors declare no conflict of interest.

## Appendix A. The Appendix Contains a Form for Recording Children Data with Intellectual Disabilities

**Form A1. Initial data form for children with intellectual disabilities.**

| M | Name | Age | IQ Level | Educational Level |
|---|------|-----|----------|-------------------|
| 1 | | | | |
| 2 | | | | |
| 3 | | | | |
| 4 | | | | |
| 5 | | | | |
| 6 | | | | |
| 7 | | | | |
| 8 | | | | |
| 9 | | | | |
| 10 | | | | |
| 11 | | | | |
| 12 | | | | |
| 13 | | | | |
| 14 | | | | |
| 15 | | | | |
| 16 | | | | |
| 17 | | | | |
| 18 | | | | |
| 19 | | | | |
| 20 | | | | |

**Appendix B. The Appendix Contains a Form for the Anger Scale in Its Initial and Final Form**

**Form A2. Anger scale in its initial form.**
Name Optional: Gender: Male/Female
Age: Study Level:
IQ Coefficient:
**If any of my colleagues make me angry, I:**

| M | Paragraph | All the Time | Once in a While | Rarely |
|---|-----------|--------------|-----------------|--------|
| 1 | I get annoyed (angry) with him. | | | |
| 2 | I cry quickly. | | | |
| 3 | I hit anything in front of me violently. | | | |
| 4 | I throw what I have on the floor. | | | |
| 5 | I curse him. | | | |
| 6 | Hit him. | | | |
| 7 | I don't talk to him. | | | |
| 8 | I withdraw from the playset. | | | |
| 9 | I break things in front of me. | | | |
| 10 | I ruin the play on my teammates. | | | |
| 11 | I push him into the ground. | | | |
| 12 | Tighten his hair. | | | |
| 13 | I feel confused. | | | |
| 14 | I can't concentrate on any other activity. | | | |
| 15 | I ruin the things of those who angered me. | | | |
| 16 | Break their toys. | | | |
| 17 | I hit myself against the wall. | | | |
| 18 | I bite my hand. | | | |
| 19 | I yell and scream. | | | |
| 20 | I tolerate it. | | | |
| 21 | I reproach him for what he did and continue playing with him. | | | |
| 22 | Continue the play (activity) calmly. | | | |
| 23 | Assault him by biting. | | | |
| 24 | I yell at him. | | | |

**Appendix C. The Appendix Contains a Form for Recreational Behavioural Programme Sessions**

**Form A3. Anger scale in its final form.**
Name Optional: Gender: Male/Female
Age: Study Level:
IQ Coefficient:

**If any of my colleagues make me angry, I:**

| M | Paragraph | All the Time | Once in a While | Rarely |
|---|---|---|---|---|
| 1 | I get annoyed (angry) with him. | | | |
| 2 | I cry quickly. | | | |
| 3 | I hit anything in front of me violently. | | | |
| 4 | I throw what I have on the floor. | | | |
| 5 | I curse him. | | | |
| 6 | Hit him. | | | |
| 7 | I don't talk to him. | | | |
| 8 | I withdraw from the playset. | | | |
| 9 | I break things in front of me. | | | |
| 10 | I ruin the play on my teammates. | | | |
| 11 | I push him into the ground. | | | |
| 12 | Tighten his hair. | | | |
| 13 | I feel confused. | | | |
| 14 | I can't concentrate on any other activity. | | | |
| 15 | I ruin the things of those who angered me. | | | |
| 16 | Break their toys. | | | |
| 17 | I hit myself against the wall. | | | |
| 18 | I bite my hand. | | | |
| 19 | I yell and scream. | | | |
| 20 | I tolerate it. | | | |
| 21 | I reproach him for what he did and continue playing with him. | | | |
| 22 | Continue the play (activity) calmly. | | | |

**Table A1.** Recreational Behavioral Program Sessions.

| Session Number | Session Topic | Session Duration | Techniques and Activities Used | Number of Sessions |
|---|---|---|---|---|
| The first | Acquaintance and introduction to the program. | 30 min | Strengthening | 1 |
| Second | Definition of anger andwhy get angry? (causes of anger). | 30 min | Reinforcement and modeling | 1 |
| Third and fourth | The consequences of anger andits impact on others' acceptance of us | 30 min | Augmentation, modeling and role playing | 2 |
| Fifth and sixth | How do I control the emotion of anger? | 30 min | Augmentation, modeling and role playing | 2 |
| Seventh – Eleventh | Apply the response cost action | 30 min | Boosting, modeling, role playing, and response cost | 5 |
| Twelfth | Recreational sports activity (football) | 30 min | Reinforcement, Football | 1 |
| Thirteenth | Recreational sports activity (basketball) | 30 min | Reinforcement, basketball | 1 |
| Fourteenth | Recreational sports activity (tug of war) | 30 min | Reinforcement, tug of war | 1 |
| Fifteenth | Recreational sports charts (jumping inside hoops) | 30 min | Boosting, jumping inside hoops | 1 |
| Sixteenth | Closing Session | 30 min | Strengthening | 1 |

**Session Ten**

**Session topic:** Response cost procedure (stage of withdrawal of boosters or fine).

**Session Objectives:**

1    Reduce anger in children by applying the response cost procedure (reinforcer withdrawal phase or fine).

**Techniques and tools used:**

Cost of response, modeling, reinforcement, Basel.

**Procedures followed when executing the session:**

**First:** At the beginning of the session, the researchers welcomed the children members of the experimental group and confirmed the attendance of everyone, and thanked them for the commitment to attend the counseling session on time, and what was done in the previous session was reviewed.

**Second:** The researchers presented to children the topic of the current session, which is to complete the reduction of anger emotion in children by applying the response cost procedure (the stage of withdrawing reinforcers or fines). This procedure aims to reduce the emotion of anger in children.

**Third:** The researchers presented the model of the response cost procedure on children, through the following steps:

The child is given a quantity of reinforcers (sticky stars in front of his picture in the painting on which his picture is placed for this purpose (the painting is placed on the wall in front of the children in class; provided that these stars are replaced at the end of the session with a reward (something the child likes), noting that whenever a quantity of reinforcers is withdrawn from him, the value of the reward will decrease as a result of performing unwanted behavior (anger emotion).

After that, the following actions are applied:

1    Identify the behavior to be modified, which is the treatment of anger emotion.
2    The time period for applying this procedure is determined (during the time period of the counseling session)
3    After giving the child to the reinforcers.
4    The child is offered an activity (assembling shapes or puzzles) and we ask him to focus on the activity and complete it to the end with help provided.
5    The child is monitored throughout the counseling session.
6    When the behavior (anger emotion) appears, a quantity of reinforcers is withdrawn, explaining the reason for this to the child as feedback.
7    We continue this procedure until the end of the session.
8    At the end of the session, we calculate the number of stars left for the child, and if he has reinforcers (stars), we equalize them with the appropriate reward.
9    We then offer the child the appropriate reward for the number of boosters (stars) he has, explaining why the value of the reward is decreasing.

**Fourth:** Review what was done in the session and its purpose, and why was the amount of enhancers withdrawn in each child? Theanswer is due to the emergence of anger in their dealings with their colleagues or during the exercise of the activity required of them, with an emphasis on the importance of reversing these behaviors, so that our relations are good with others, and so that they accept us, and there is love from others for us, in addition to retaining the rewards and reinforcements that we have.

**Fifth**: At the end of the session, the researchers thanked the children members of the experimental group for their good participation and interaction in the session, and reminded himof the date of the next session.

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
