# Peer review of "The Effectiveness of a Recreational Behavioural Programme in Reducing Anger among Children with Intellectual Disabilities at the Primary Stage"

_ejihpe, doi:10.3390/ejihpe13060072_

Round 1
Reviewer 1 Report
Your work I am sure is valuable. I need a better understanding and cleaning up of your manuscript to be more helpful. I hope my comments help with overall cleaning and clarity.
Line 15, you indicate 20 children, but your two groups add up to 24.
Line 16-17, I do not understand Number written four times.
Line 21, Please provide effect size values for the results. I see reading your manuscript you indicated r (line 227). I do not see r’s in your tables but R and they are > at times than 1.
Should intellectual disabilities be IDs? Is this standard? ADD seems standard, but I wonder if ID is.
Line 91, It seems the e.g., should be in the brackets.
Line 112, The next large block of words seems to need formatting and more paragraphs to break it up.
Line 144, This is a run on sentence of sorts.
Line 149, Same issue of 20 participants but two groups of 12.
Line 151, The Number issue appears again.
2.3. Assessments, a) and then the B) are not consistent. This section needs a couple of paragraphs to break it up.
Line 248, The spacing and commas need work.
You are not consistent with your p= or P= in your results.
Figure 1, I do not understand the percentage line.
I do not understand the letter superscripts in the tables. I do not understand your tables in general.
The conclusions section needs paragraphs.
Hello,
The manuscript needs cleaned up. English is fine.
Author Response
Response to Reviewer 1
Your work I am sure is valuable. I need a better understanding and cleaning up of your manuscript to be more helpful. I hope my comments help with overall cleaning and clarity.
Thank you for your comments, which added to the authors a lot of valuable information that contributed significantly to raising the value of the research.
Line 15, you indicate 20 children, but your two groups add up to 24.
Response: the Changes have been made
Line 16-17, I do not understand the Number written four times.
Response: the Changes have been made
Line 21, Please provide effect size values for the results. I see reading your manuscript you indicated r (line 227). I do not see r’s in your tables but R and they are > at times than 1.
Response: the Changes have been made
Should intellectual disabilities be IDs? Is this standard? ADD seems standard, but I wonder if ID is.
Response: It was applied to children with Down syndrome
Line 91, It seems the e.g., should be in the brackets.
Line 112, The next large block of words seems to need formatting and more paragraphs to break it up.
Response: the Changes have been made
Line 144, This is a run on sentence of sorts.
Response: the Changes have been made
Line 149, Same issue of 20 participants but two groups of 12.
Response: the Changes have been made
Line 151, The Number issue appears again.
Response: the Changes have been made
2.3. Assessments, a) and then the B) are not consistent. This section needs a couple of paragraphs to break it up.
Response: the Changes have been made
Line 248, The spacing and commas need work.
You are not consistent with your p= or P= in your results.
Response: the Changes have been made
Figure 1, I do not understand the percentage line.
Response: The figure shows the improvement rates of the variables of the anger scale between the pre-measurement and the post-measurement of the experimental group, where it was found to improve in different proportions.
I do not understand the letter superscripts in the tables. I do not understand your tables in general.
Response: Changes were made where the average ranks and total ranks were deleted and the mean and standard deviation were added making the result more visible to readers.
The conclusions section needs paragraphs.
Response: the Changes have been made
Comments on the Quality of English Language
Hello,....The manuscript needs cleaned up. English is fine.
Response : Modifications have been made, This document certifies that the manuscript listed below was edited by Cambridge Proofreading LLC for English grammar, punctuation, spelling,
https://drive.google.com/file/d/10offH0x8MUOJaFOM8S3Pp01Ge5MEkR-7/view?usp=share_link
In the end, the authors thank you for your comments, which contributed greatly to increasing the experiences and information of the authors and clarifying the whole picture in front of them, and also your interest in the manuscript, and which contributed to producing the study in a good way. we wish you more progress and sophistication and enjoy health and wellness.
Thank you so much
Reviewer 2 Report
Interesting, and good subject
Author Response
Response to Reviewer 2
We offer you the highest verses of love, appreciation and gratitude, for all that you have provided, you are one of those who carry the honesty of work and sincerity in their necks and are keen to provide everything that is new, wonderful and useful, if at the expense of themselves, thank you for your continuous support, and your comment is a medal that we are proud of, you thank you very much for your comments that make us move forward, and we wish you more progress and sophistication and enjoy health and wellness.
Thank you so much

Reviewer 3 Report
Overview
The authors aimed to investigate the impact of a recreational behavioural program to reduce the degree of anger among children with intellectual disabilities in the primary stage.
The study is interesting; however, it needs to be reviewed by a native English speaker experienced in scientific manuscript writing.
The formatting of the manuscript is wrong, and this makes it difficult to read.
The manuscript needs to be significantly improved to be considered for publication.
Below are my comments.
Specific comments
Abstract
The abstract does not clearly summarize the various sections of the manuscript. Furthermore, the text should be perfected using scientific writing.
Replace with: “This study aimed to investigate the impact of a recreational behavioral program on reducing the degree of anger among children with intellectual disabilities (ID) in the primary stage”.
Replace with “The study was implemented with 20 children who were randomly divided into two groups”.
Replace “Number” with “Scores”.
What do you mean by "homogeneity"? Clarify what differences.
It is not clear what tests were used to measure the dependent variables.
Keywords
Keywords must be different from the words in the title.
Introduction
The introduction has some typos. It is written in a confusing way. I recommend shortening it. Follow the logical thread: what is known, what is unknown, aims and research hypotheses. No more than 700/800 words. The introduction is not an exhaustive review. The background must be synthetic.
Methods
This section is also too confusing. The section has some typos.
Replace "Experimental approach to the problem" with "Participants and Study Design".
The recreational behavioral program is an intervention. It is not an evaluation tool. It is not the dependent variable. Why is it included in the "Assessments" paragraph? It must have a separate paragraph.
Lies 225-234: Please insert a subparagraph "Statistical analysis".
Results
Line 234. The results should be summarized and clarified. This section is also too confusing.
The tables must first be indicated in the text and then immediately inserted. Check the Table 2.
Discussion
Study limitations were not discussed. Add to the end of the discussion by emphasizing the strengths and limitations of the study.
Conclusions
The conclusions must be summarized. There must be no data.
The following must be written clearly and concisely: 1) Your main findings (i.e., the take-home message); 2) Future directions/implications.
Remember to restate the novelty of the paper according to the current literature to help better readers understand how this paper is different from others already published.
Dear authors,
the study is interesting; however, it needs to be reviewed by a native English speaker experienced in scientific manuscript writing.
The formatting of the manuscript is wrong, and this makes it difficult to read.
Author Response
Response to Reviewer 3
Abstract
The abstract does not clearly summarize the various sections of the manuscript. Furthermore, the text should be perfected using scientific writing.
Response: the Changes have been made
Replace with: “This study aimed to investigate the impact of a recreational behavioral program on reducing the degree of anger among children with intellectual disabilities (ID) in the primary stage”.
Response: the Changes have been made
Replace with “The study was implemented with 20 children who were randomly divided into two groups”.
Response: the Changes have been made
Replace “Number” with “Scores”.
Response: the Changes have been made
What do you mean by "homogeneity"? Clarify what differences.
Response:The purpose of homogeneity is to show that samples are symmetrical and uniform
It is not clear what tests were used to measure the dependent variables.
Response: It was clarified inside the manuscript
Keywords
Keywords must be different from the words in the title.
Response: the Changes have been made
Introduction
The introduction has some typos. It is written in a confusing way. I recommend shortening it. Follow the logical thread: what is known, what is unknown, aims and research hypotheses. No more than 700/800 words. The introduction is not an exhaustive review. The background must be synthetic.
Response: the Changes have been made
Methods
This section is also too confusing. The section has some typos.
Replace "Experimental approach to the problem" with "Participants and Study Design".
The recreational behavioral program is an intervention. It is not an evaluation tool. It is not the dependent variable. Why is it included in the "Assessments" paragraph? It must have a separate paragraph.
Response: the Changes have been made
Lies 225-234: Please insert a subparagraph "Statistical analysis".
Response: the Changes have been made
Results
Line 234. The results should be summarized and clarified. This section is also too confusing.
The tables must first be indicated in the text and then immediately inserted. Check the Table 2.
Response: the Changes have been made
Discussion
Study limitations were not discussed. Add to the end of the discussion by emphasizing the strengths and limitations of the study.
Response: Study limitations and strengths of the study were discussed
Conclusions
The conclusions must be summarized. There must be no data.
The following must be written clearly and concisely: 1) Your main findings (i.e., the take-home message); 2) Future directions/implications.
Response: the Changes have been made
Comments on the Quality of English Language
Hello,....The manuscript needs cleaned up. English is fine.
Response : Modifications have been made, This document certifies that the manuscript listed below was edited by Cambridge Proofreading LLC for English grammar, punctuation, spelling,
https://drive.google.com/file/d/10offH0x8MUOJaFOM8S3Pp01Ge5MEkR-7/view?usp=share_link
In the end, we would like to thank you for this valuable information that added a lot of experiences to us and we benefited from your comments and guidance that raised the value of the manuscript, and that contributed to producing the study in a good way. your success and enjoyment of health and wellness.
Thank you so much

Round 2
Reviewer 1 Report
Hello, I appreciate your revision efforts. Your manuscript contains better words, but is still full of countless errors and inconsistencies.
Line 19, extra space after anger before the comma.
Line 21, I do not see AT, LT, and EA anywhere before now and thus the abbreviations are confusing.
Same with ASW. IQ seems to always be IQ.
Keywords need to be on the next line.
Line 36, physiological
Line 39, also
The way lines and paragraphs split you have a number of – in the words.
Line 46, externalizing, regulates, and so on. I am going to stop, but this needs lots of attention.
Line 49, need a space after the ]. There are more of these errors. I am going to stop indicating each.
Line 54, This is this.
Seems like lots of errors for a paid language writing service.
Paragraph starting on line 93 has lots of errors and is in bold. Again, lots of errors.
Paragraph starting on line 115, same thing, full of errors.
Figure 1, I do not understand the middle error bar between each of the main error bars.
Figure is inserted incorrectly.
I do not understand the dashes after the Z scores in Table 3 and the other tables.
I stopped as the manuscript still needs the basic corrections. It is too distracting to try to understand the manuscript with so many errors/typos/inconsistencies. It makes me wonder about the data analyses.
Words are fine, the errors and typos seem more than in the first version. The authors are not performing any quality check on their end. This is very disappointing.
Author Response
Review Response Form(1)
*Line 19, extra space after anger before the comma.
Response: Changes have been made
*Line 21, I do not see AT, LT, and EA anywhere before now and thus the abbreviations are confusing.
Response: Changes have been made and the meaning of the abbreviation has been clarified
*Same with ASW. IQ seems to always be IQ.
Response: Changes have been made and the meaning of the abbreviation has been clarified
*Keywords need to be on the next line.
Response: Changes have been made
*Line 36, physiological
Response: Changes have been made
*Line 39, also
Response: Changes have been made
*The way lines and paragraphs split you have a number of – in the words.
Response: Changes have been made
*Line 46, externalizing, regulates, and so on. I am going to stop, but this needs lots of attention.
Response: Changes have been made
*Line 49, need a space after the ]. There are more of these errors. I am going to stop indicating each.
Response: Changes have been made
*Line 54, This is this.
Response: Changes have been made
*Seems like lots of errors for a paid language writing service.
Response: Changes have been made, and the English language was reviewed throughout the entire research process.
*Paragraph starting on line 93 has lots of errors and is in bold. Again, lots of errors.
Response: Changes have been made, and the English language was reviewed throughout the entire research process.
*Paragraph starting on line 115, same thing, full of errors.
Response: Changes have been made, and the English language was reviewed throughout the entire research process.
*Figure 1, I do not understand the middle error bar between each of the main error bars.
Response: Changes have been made and error bars have been removed.
*Figure is inserted incorrectly.
Response: Changes have been made
*I do not understand the dashes after the Z scores in Table 3 and the other tables.
Response: Changes have been made
*I stopped as the manuscript still needs the basic corrections. It is too distracting to try to understand the manuscript with so many errors/typos/inconsistencies. It makes me wonder about the data analyses.
.
Comments on the Quality of English Language
Words are fine, the errors and typos seem more than in the first version. The authors are not performing any quality check on their end. This is very disappointing.
We apologize for exhausting you. The study has drawn attention and had the English language reviewed. We truly appreciate your interest in the study and the care with which you offered your ideas and observations, which helped to make the study's primary themes more clear, pertinent, and important. We hope you'll like the adjustments we've made.
https://drive.google.com/file/d/1U1bU0hi5gvONiPLsV7yoqybBNhQ_McHW/view?usp=drive_link
thank you

Reviewer 3 Report
Dear authors,
thank you for your replies.
The changes made have greatly improved the manuscript.
I ask you again to insert a new subparagraph in the Methods section.
After the paragraph "2.3 Assessments" and "2.3.1 Testing Procedures" add the paragraph "2.4 Recreational behavioral program" which contains lines 166 to 184 and Table 1.
No comment.
Author Response
Review Response Form(3)
I ask you again to insert a new subparagraph in the Methods section.
After the paragraph "2.3 Assessments" and "2.3.1 Testing Procedures" add the paragraph "2.4 Recreational behavioral program" which contains lines 166 to 184 and Table 1.
We have been adjusting. We're sorry if this has worn you out. We sincerely thank you for your interest in the study and for the thoughtfulness with which you shared your thoughts and observations, which improved the clarity, applicability, and significance of the study's main themes. The changes we made are ones we think you'll like.
thank you
